deepAMPNet: a novel antimicrobial peptide predictor employing AlphaFold2 predicted structures and a bi-directional long short-term memory protein language model

Zhao Fei
Qiu Junhui
Xiang Dongyou
Jiao Pengrui
Cao Yu
Xu Qingrui
Qiao Dairong
Xu Hui xuhui_scu@scu.edu.cn
Cao Yi cyi@scu.edu.cn
Microbiology and Metabolic Engineering Laboratory of Sichuan Province, College of Life Science, Sichuan University , Chengdu , Sichuan , China
Date Swapneeta
Electronic publication date: 2024 Jul 19
Publication date: 2024
Volume: 12
Electronic Location ID: e17729
Received 2024 Feb 28; Accepted 2024 Jun 20
Copyright: ©2024 Zhao et al.
Copyright year: 2024
Copyright holder: Zhao et al.
License: This is an open access article distributed under the terms of the Creative Commons Attribution License, which permits unrestricted use, distribution, reproduction and adaptation in any medium and for any purpose provided that it is properly attributed. For attribution, the original author(s), title, publication source (PeerJ) and either DOI or URL of the article must be cited.
License URL: https://creativecommons.org/licenses/by/4.0/

Keywords: Bioinformatics, Antimicrobial peptide, Graph neural network, Bi-LSTM, Computational biology, Protein identification

Funding: National Natural Science Foundation of China 32071479 32271535 32171473 Department of Science and Technology of Sichuan Province 2022NSFSC0243 This work was supported by National Natural Science Foundation of China (32071479, 32271535, 32171473), Department of Science and Technology of Sichuan Province (2022NSFSC0243). The funders had no role in study design, data collection and analysis, decision to publish, or preparation of the manuscript.

==============================
Background

Global public health is seriously threatened by the escalating issue of antimicrobial resistance (AMR). Antimicrobial peptides (AMPs), pivotal components of the innate immune system, have emerged as a potent solution to AMR due to their therapeutic potential. Employing computational methodologies for the prompt recognition of these antimicrobial peptides indeed unlocks fresh perspectives, thereby potentially revolutionizing antimicrobial drug development.

Methods

In this study, we have developed a model named as deepAMPNet. This model, which leverages graph neural networks, excels at the swift identification of AMPs. It employs structures of antimicrobial peptides predicted by AlphaFold2, encodes residue-level features through a bi-directional long short-term memory (Bi-LSTM) protein language model, and constructs adjacency matrices anchored on amino acids’ contact maps.

Results

In a comparative study with other state-of-the-art AMP predictors on two external independent test datasets, deepAMPNet outperformed in accuracy. Furthermore, in terms of commonly accepted evaluation matrices such as AUC, Mcc, sensitivity, and specificity, deepAMPNet achieved the highest or highly comparable performances against other predictors.

Conclusion

deepAMPNet interweaves both structural and sequence information of AMPs, stands as a high-performance identification model that propels the evolution and design in antimicrobial peptide pharmaceuticals. The data and code utilized in this study can be accessed at https://github.com/Iseeu233/deepAMPNet.

Introduction

Antimicrobial resistance (AMR), a consequence of antibiotic misuse, is snowballing into a formidable health challenge for humanity (O’Neill, 2016). This resistance emerges when bacteria undergo evolution rendering them invulnerable to conventional treatments (Murray et al., 2022). Regarded as crucial components of the innate immune defense, antimicrobial peptides (AMPs) possess broad-spectrum potency against a variety of harmful microorganisms, inclusive but not limited to bacteria, fungi, and viruses (Ahmed et al., 2019). AMPs wield their striking mechanism of influence by selectively binding to the membrane structure of pathogenic microorganisms, causing a disruption; or by direct micropore formation on the cell membranes initiating an outflow of cellular contents, thereby eliminating the pathogenic entities (Steinstraesser et al., 2011). This approach not only amplifies their antimicrobial efficiency but also significantly diminishes the opportunity for drug resistance to evolve. AMPs are gaining recognition as potent substitutes to traditional therapies (Lazzaro, Zasloff & Rolff, 2020). Consequently, unearthing new AMPs and fostering the evolution of innovative AMP-based drugs is of paramount importance to counteract antimicrobial resistance.

Conventional wet-lab experiments for identifying AMPs can be intricate and require notable time investments, proving insufficient for high-throughput identification (Medema & Fischbach, 2015). The computational identification of AMPs has ascended to the forefront as a mainstream approach, and a number of comprehensive antimicrobial peptide databases have been established (Bin Hafeez et al., 2021). The ample utilization of artificial intelligence methods for AMPs identification spans two principal categories: traditional machine learning techniques like support vector machine (SVM), random forest (RF), and artificial neural network (ANN) (Xu et al., 2021). The CAMP3 database (Waghu et al., 2016) website offers a tailored tool for AMPs identification, featuring several machine learning methods to choose from; Fingerhut et al. (2021) developed SVM-based ampir for AMPs identification; AMPfun developed by Chung et al. (2019) supports the identification of AMPs using decision tree (DT), RF, and SVM; Emerging deep learning methods such as convolutional neural network (CNN), long short-term memory (LSTM), bidirectional encoder representations from transformers (BERT), etc. are also used in the field of AMPs identification (García-Jacas et al., 2022). Ma et al. (2022) combined three natural language processing models (NLP), LSTM, Attention, and BERT, creating a unified pipeline adept at identifying potential AMPs from human gut microbiome; Zhou et al. (2023) spearheaded the creation of a model known as Tri-net, pinpointing three signature features to capture peptide information contained in serial fingerprints, sequence evolutions, and physicochemical properties. Every feature is input into parallel modules to enhance the precise prediction of anticancer peptides (ACPs) and AMPs; Wang et al. (2023) used enhanced bi-directional LSTM (EBiLSTM) model to construct an AMPs predictor with binary profile feature (BPF) and pseudo amino acid composition (PseAAC) for effective localized sequence capture and amino acid information extraction.

Beyond choosing dedicated algorithms for AMPs identification, many researchers showed a propensity for feature engineer using different protein features (Xu et al., 2021). Predominantly, they utilized: amino acid composition (AAC) (Meher et al., 2017), dipeptide composition (DPC) (Gautam et al., 2013), pseudo amino acid composition (PseAAC) (Chou, 2011), adaptive skip DPC (ASDC) (Wei, Tang & Zou, 2017), PSSM profile (Fu et al., 2020), structural features (α-Helix, β-Sheet) (Lin et al., 2019), physicochemical properties (Lin et al., 2021) and so on. Lin et al. (2021) recorded 115 physicochemical properties of amino acids using the R package Peptides. And hierarchical clustering came into play for the final selection of six essential physicochemical properties for encoding amino acid sequences. Subsequently, these were leveraged to architect a model for identifying AMPs. Shaon et al. (2023) built AMP-RNNpro, a recurrent neural network (RNN)-based model devised for AMPs prediction. This model relied on eight feature encoding methods, diligently chosen based on four key criteria -amino acid composition, autocorrelation, and pseudo amino acid composition. These potent combinations of features allowed efficient computational identification of AMPs.

As a noteworthy point, during traditional protein characterization engineering involving antimicrobial peptide sequences of diverse lengths, it is necessary to supplement with either 0 or ‘X’ at the terminus in order to assure input dimension uniformity (Xu et al., 2021). Interestingly, the graph neural network (GNN) has been garnering considerable interest in AMPs characterization, thanks to its non-reliance on consistency of input dimensions and top-notch performance (Fernandes et al., 2023). Ruiz Puentes et al. (2022) developed AMPs-Net, ingeniously transformed the peptide sequence into a graph representation, each amino acid node was depicted using nine physicochemical properties, and the edge between amino acid nodes was determined with three distinct properties. Yan et al. (2023) proposed a graph attention network (GAT) model for the identification of antimicrobial peptides, named sAMPpred-GAT. This model made use of one-hot features, position features, PSSM and HMM features of proteins for representing amino acids. Furthermore, Cα distances in protein structures were employed to depict the edges between the amino acid nodes.

Identification of antimicrobial activity of existing proteins has become essential for mining novel antimicrobial peptides. Table 1 summarizes 14 representative AMP identification models compared in this study according to their algorithm, feature selection and publication year. The analysis and comparison of the existing models of AMPs revealed the following limitations: (1) Lack of global information of the sequence. Most of the existing methods have the features of amino acid sequences extracted one by one during feature extraction (Xu et al., 2021), missing the global information of the sequences; (2) redundant or insufficient number of features. The number of features used by the existing model may be insufficient or heavily redundant. For example, some methods directly select a few physicochemical properties from hundreds of physicochemical properties (Lin et al., 2021), which may lead to serious redundancy and low quality of features. While some methods only use secondary structure features of proteins (e.g.,  α-Helix, β-Sheets) (Fernandes, Rigden & Franco, 2012; Beltran, Aguilera-Mendoza & Brizuela, 2018; Vishnepolsky & Pirtskhalava, 2014) or one-hot coding of protein sequences, which may lead to insufficient features to be difficult to learn. (3) Inappropriate neural networks. Some models fail to design specific deep learning algorithms based on the characteristics of different features, or even using the same or similar neural networks when processing different features (Lobanov et al., 2023). (4) Insufficiency of training data. A portion of the models were trained with only a small number of thousands of AMPs, which may be insufficient for an AMPs family from different species with different function. Due to the above shortcomings, the existing models with high false positive rate, insufficient generalization ability, have good performance only in specific datasets. Clearly, there still exists significant potential for optimization.

Table 1 Statistical information of the compared models.

Method	Algorithm	Feature selection	Reference	
ADAM-HMM	HMM	secondary structure	Lee et al. (2015)	
CAMP3	ANN, RF, SVM, DA	AAC	Waghu et al. (2016)	
iAMPpred	SVM	AAC, Physicochemical properties, Structure features	Meher et al. (2017)	
AMPEP	RF	AAC	Bhadra et al. (2018)	
AMPScannerV2	LSTM, CNN	AAC	Veltri, Kamath & Shehu (2018)	
AMPfun	DT, SVM, RF	AAC, Physicochemical properties	Chung et al. (2019)	
AmpGram	RF	AAC	Burdukiewicz et al. (2020)	
Deep-AMPEP30	CNN, SVM, RF	AAC	Yan et al. (2020)	
ampir	SVM	AAC	Fingerhut et al. (2021)	
AI4AMP	SVM, RF, NLP	AAC, Physicochemical properties	Lin et al. (2021)	
amPEPpy	RF	Physicochemical properties	Lawrence et al. (2021)	
AMPlify	Bi-LSTM	AAC	Li et al. (2022)	
AMPpred-MFA	LSTM, CNN	AAC	Li et al. (2023)	
sAMPpred-GAT	GAT	AAC, Structure features, Evolutionary features	Yan et al. (2023)	
Notes.

Full names of the algorithms and features HMM hidden Markov model

ANN artificial neural network

RF random forest

SVM support vector machine

DA discriminant analysis

LSTM long short-term memory

CNN convolutional neural networks

Bi-LSTM bidirectional LSTM

DT decision tree

NLP natural language processing

GAT Graph Attention Network

AAC amino acid composition

The advent of Alphafold2 (Jumper et al., 2021) has paved the way for accurate protein structure prediction, with over 200 million protein structures already predicted in the AlphaFold protein structure database (Varadi et al., 2022). Previous research indicated that these predicted protein structures can be harnessed for downstream functional analyses (Koehler Leman et al., 2023). This opens up the exciting possibility of using antimicrobial peptide structures for the identification of AMPs. Consequently, in order to address the shortcomings of existing methods, we present a novel model named deepAMPNet in this study. The structure of deepAMPNet is visually represented in Fig. 1. It harnessed a hierarchical pool graph convolutional neural network built upon structure learning to devise a classification model. This model was equipped for recognizing AMPs by encoding residue-level features via bi-directional LSTM, utilized the antimicrobial peptide structures predicted by AlphaFold2. Differing from existing models, deepAMPNet leverages protein language models to capture global information of sequences, models the predicted tertiary structures as graph data to conform to the input requirements of graph neural network (GNN), mitigates potential feature redundancy or inadequacy by comparing the performance of various feature quantities, and collects data from five comprehensive AMP databases to ensure generalization capability. Tested on two external independent datasets, deepAMPNet not only triumphed over other state-of-the-art predictors in accuracy, but also excelled in other metrics such as AUC, MCC, and precision, matched or surpassed the performance of other predictors. To a certain extent, deepAMPNet addressed the limitations of existing models, and can expedite the acquisition of candidate sequences for antimicrobial peptide drug development, while also can offer insights and inspiration for the intelligent prediction of other functional proteins.

Figure 1 The framework of deepAMPNet.

The protein sequence is channeled through Bi-LSTM model to obtain residue descriptors. A contact map is then constructed considering the distances among protein residues. Then, the contact map and residue-level descriptors are regarded as the adjacency matrix and feature matrix, respectively, that are used to supply information into the GCN network for AMPs recognition.

Materials & Methods

Independent test datasets

To achieve comparable performance evaluation, we incorporated two external independent test datasets in this study. As delineated in Table 2, these datasets are named as XUAMP and MFA_test. XUAMP, proposed by Xu et al. (2021), unified AMPs from an array of all-inclusive public peptide databases. It eliminated sequences previously utilized as training data in evaluated models, and retained only sequences with lengths between 10 and 100. Homology bias and redundancy were eradicated using cd-hit. The non-antimicrobial peptides (Non-AMPs) was reaped from Uniprot, processed similarly, and randomly chosen, resulting in a balance of 1,536 AMPs and 1,536 Non-AMPs. MFA_test, presented by Li et al. (2023), was similarly treated and contained 615 AMPs and 606 Non-AMPs.

Table 2 Datasets and statistical information.

Dataset	Positive (AMPs)	Negative (Non-AMPs)	Total	
XUAMP	1,536	1,536	3,072	
MFA_test	615	606	1,221	
Training dataset	11,485	11,485	22,970	

Training dataset

The AMPs used in this study were sourced from five databases: APD3 (Wang, Li & Wang, 2016), DBAASP (Pirtskhalava et al., 2021), dbAMP (Jhong et al., 2022), DRAMP (Shi et al., 2022), and LAMP2 (Ye et al., 2020). Integrating all the data from these databases resulted in a grand total of 60,218 redundant AMPs. The duplicates were eliminated, retaining only the antimicrobial peptides with lengths spanning from 6–200. And sequences containing non-canonical residues (like ‘B’, ‘J’, ‘O’, ‘U’, ‘X’ or ‘Z’) are removed because their structures are not predictable by AlphaFold2. Sequence comparisons were conducted by using blastP (Boratyn et al., 2012), and 5,954 AMPs’ structures were procured by downloading from the AlphaFold protein structure database. An aggregate of 5,531 AMPs’ structures were predicted locally using the AlphaFold2. This resulted in a comprehensive collection of 11,485 AMPs as a positive dataset.

The negative dataset was constructed from the Uniprot (http://www.uniprot.org) by excluding keywords such as ‘antimicrobial’, ‘antibacterial’, ‘antifungal’, ‘anticancer’, ‘antiviral’, ‘antiparasitic’, ‘antibiotic’, ‘antibiofilm’, ‘effector’, and ‘excreted’. We set the length range from 6–200 and processed sequences similarly to AMPs, eradicating non-canonical residues. Sequences possessing more than 40% similarity to any AMPs, and sequences with over 40% similarity amongst Non-AMPs, were both discounted. Out of the remaining sequences, we randomly picked 11,485 data points to constitute the Non-AMPs dataset, and all structures were downloaded from the AlphaFold protein structure database.

The length distribution of the data within the training dataset is visually represented in Fig. 2, while the distribution of amino acids in AMPs and Non-AMPs is shown in Fig. 3 and Fig. S1.

Note that in order to accurately and objectively assess the performance of deepAMPNet, we used cd-hit program (Fu et al., 2012) to remove sequences in the training dataset that have more than 90% similarity to those in XUAMP and MFA_test, respectively, when performing model training.

AlphaFold2 benchmark dataset

The AlphaFold structure database only predicts sequences with unidentified structures; hence, sequences with known structures from the PDB were not predicted (Varadi et al., 2022). From the annotations of the DBAASP database for AMPs, PDB sequence numbers associated with certain AMPs were obtained, totaling 339 data entries. Thus, the 339 AMPs underwent structural prediction using AlphaFold2. The actual structures in the PDB were downloaded for an ensuing assessment of AlphaFold2’s performance in predicting antimicrobial peptide structures.

Framework of deepAMPNet

The framework of deepAMPNet is shown in Fig. 1. The structure of deepAMPNet encompasses two fully-connected layers aimed at reducing node feature dimensions, three layers dedicated to graph convolution, two layers for hierarchical graph pooling with structure learning (HGPSL), and two dropout layers to avert overfitting. Finally, it consists of three fully-connected layers deployed for binary classification.

Figure 2 The length distribution of training dataset.

Figure 3 Amino acid distribution of AMPs and Non-AMPs of training dataset.

Graph convolution layer

To improve the representation of the three-dimensional structure and physicochemical properties of amino acids, AMPs can be modeled as a graph based on its structure. Graph data is characterized by two types of features: nodes and edges. Leveraging the sequence and structural information of proteins, their structure is represented as graph data that can be recognized by computers. This is accomplished by forming residue descriptors of proteins to depict the node feature matrix, and by utilizing contact maps - indicative of residue interaction - to represent the adjacency matrix.

Graph Convolution (Bronstein et al., 2017) as a deep learning algorithm for demonstrating reliable performance on graph data, the input requirements are: node feature matrix and adjacency matrix. Any graph convolution layer can be presented as a nonlinear function: (1) Hl+1=σD ~−12A ~D ~−12HlWl

where Hl+1 denotes the node feature matrix of layer l+1, A ~∈RN×N=A+I means the graph adjacency matrix with self-loops, D ~∈RN×N is the diagonal degree matrix of A ~. The degree of a node is actually the number of neighbors connected to that node. Hl means the graph node feature matrix in the l-th layer. In this study, for the input layer, l = 0, Hl means the input residue-level features. Wl means the trainable weight matrix for the layer l+1, and σrepresents the nonlinear activation function, which is ReLU in this study.

Bi-LSTM model for encoding residue-level features

Protein language model serves as a powerful deep learning method for extracting information from protein sequences. From readily available sequence data alone, protein language model has been able to reveal evolutionary, structural and functional features of proteins (Bepler & Berger, 2021). The pre-trained bi-directional long short-term memory (Bi-LSTM) model proposed by Bepler & Berger (2021) was used to represent residue-level features of proteins, which was specialized for protein sequence coding and can be used for transfer learning for downstream protein functional analysis. The Bi-LSTM was trained with 76,215,872 protein sequences derived from UniRef90, while 28,010 protein sequences with known structures from SCOP were employed for structurally supervised learning. The entire process spanned 51 days on a single NVIDIA V100 GPU.

The Bi-LSTM model leveraged the protein sequence, encoded via 21-dimensional one-hot encoding (20 standard amino acids + 1 unknown amino acid), as the input. This model boasted a three-layered bi-directional long short-term memory network with 1,024 hidden neurons in each layer. The protein’s residue-level features stem from the hidden layer of the LSTM model, and the output feature dimensions total 6,165. Of these dimensions, the first 21 pertain to the one-hot representation of the residues. Additionally, the Bi-LSTM model includes a fully connected layer capable of encoding the residues into 100-dimensional vectors.

Owing to the substantial computational resources needed to compute the residue’s coded feature dimension 6,165, the initial 21 dimensions of the residue, along with the features succeeding the 21 dimensions, were dimensionally constrained by two fully connected layers respectively, prior to directing the output to the graph convolution layer: (2) HLM=onehot+1024×n×2

(3) H1=σHLM:21⋅W1+b1

(4) H2=σHLM21:⋅W2+b2

(5) Hinput=H1,H2

where HLM represents the residue characteristics obtained by Bi-LSTM model, n is the number of layers of the bidirectional LSTM, W1, W2, b1 and b2are the weights that can be trained in the graph convolution layer.

In this study, we further probed the influence of different quantities of residue-level features on the deepAMPNet performance, considering 21 dimensions (one-hot), 2,069 dimensions (one-hot + first Bi-LSTM layer), 4,117 dimensions (one-hot + first 2 Bi-LSTM layers), 6,165 dimensions (one-hot + 3 Bi-LSTM layers), and a concise 100 dimensions (output of the Bi-LSTM’s fully-connected layer).

Hierarchical graph pooling with structure learning

Hierarchical graph pooling with structure learning (HGPSL) (Zhang et al., 2019) is a novel graph classification method with complex pooling operations. The method utilizes node features and topological information of the entire graph to predict the label associated with the graph. A usual approach to graph classification involves aggregating all node data to formulate a universal representation of the graph. This practice overlooks the entire graph’s structural data. Conversely, graph neural network (GNN) is incapable of hierarchically agglomerating node information. Furthermore, sometimes it’s essential to assign unique significance to the graph’s substructures to create a substantial representation of the graph, as different substructures might contribute variably to the all-encompassing graph representation. HGPSL offers a graph pooling process that can learn from both local and global structures of the graph and yield hierarchical representations.

Graph pooling operation

Should a node in the graph be representable by its adjacent nodes, it can consequently be erased without any loss of information. The HGPSL pooling layer, appeares after each convolutional layer, operates as a nonparametric layer fulfilling the graph pooling task by scoring each node followed by node selection. The steps of node selection are (1) calculate the node information scores by Manhattan distance between node and its neighboring nodes: (6) p=γGi=Iik−Dik−1AikHik

where Aik denotes the adjacency matrix, Hik denotes the node representation matrix, Dik denotes the diagonal degree matrix of Aik, Iik denotes the identity matrix, and γ is the node score coding function, which in this case refers to the Manhattan distance; (2) reorder the nodes based on the information scores; (3) retain a few nodes with top scores based on the pooling rate to form the feature matrix of the next layer as well as the uncorrected adjacency matrix. With this processing, both node features and graph structure information are retained to the next layer.

Structure learning mechanism

The above graph pooling process is equivalent to constructing a subgraph of the current graph, but graph pooling does not further update the structural features of the subgraph, which may result in the loss of topological information. After the pooling operation, the subgraph structure may lead to disconnection of highly correlated nodes in the original graph, which in turn may hinder message passing and effective node feature learning. Therefore, HGPSL uses an attention-based structure learning mechanism to compute the energy E for each pair of nodes: (7) Eip,q=σa→Hikp,:∥Hik:,qT+λ⋅Aikp,q

where p, q denotes two node numbers, Hikp,:, Hik:,q denote the p-th and q-th rows of the node feature matrix Hik, respectively, and σ represents the nonlinear activation function, which is ReLU in this study. λ is the trade-off parameter between nodes, and λ⋅Aikp,q is used to preserve the original node connection information, and the sparse neighbor matrix of a new subgraph can be updated by using sparsemax through the energy E.

Fully connected layer

The linear layer (Goodfellow, Bengio & Courville, 2016) was used to convert the final output to the probability of whether or not an antimicrobial peptide, which in this study consists of three fully connected layers: (8) yi=σwxi+b

where σ denotes the activation function, log_softmax (Goodfellow, Bengio & Courville, 2016), used in this study, was used to convert the output of the fully connected layer. yi denotes the final output. w denotes the trainable weights, b denotes the bias unit, and xi denotes the output of the last fully connected layer.

Contact map

Alphafold2 stores the structure of a protein by generating a PDB file that contains the 3-dimensional coordinates for all constituent atoms of the protein. To ascertain interactions between two residues, only the distance between the paired Cα atoms is considered. By employing a distance threshold, if the distance between the pertaining Cα atoms of the two residues falls within this threshold, they’re perceived as in contact and assigned ‘1’ in the adjacency matrix. In contrast, if this distance surpasses the threshold, they’re deemed non-interactive and marked as ‘0’ in the matrix. In this study, we investigated the impact of varying distance thresholds on deepAMPNet’s performance.

Evaluation metrics

In this study, the following six metrics were used to evaluate the performance of the proposed model: (9) Accuracy=TP+TNTP+TN+FP+FN

(10) MCC=TP×TN−FP×FNTP+FNTP+FPTN+FPTN+FN

(11) Sensitivity=TPTP+FN

(12) Specificity=TNTN+FP

(13) Precision=TPTP+FP

(14) F1=2×TP2×TP+FN+FP

where TP, FP, TN and FN are the number of true positives, false positives, true negatives and false negatives, respectively. MCC denotes Matthews correlation coefficient, F1 denotes F1-score.

Results

Performance of AlphaFold2 in predicting AMPs’ structures

The performance of AlphaFold2 in predicting the structure of AMPs was evaluated using a collection of 339 AMPs with experimental structures. Briefly, performance was assessed by comparing the predicted structures with the experimental structures.

As for the evaluation metrics employed for protein structure comparison, the Cα root mean square deviation (RMSD) (Kufareva & Abagyan, 2012) of proteins was initially used. Following the comparison of the predicted structures and the experimental structures using pymol, the RMSD distribution for all structure comparisons can be observed in Fig. 4. The kernel density curve suggests its peak to be stationed around 1Å, and the histogram reflects that in excess of 70% (246/339) of the structural comparison RMSDs are less than 2 Å.

Figure 4 The distribution of RMSD for all protein structure comparisons.

AlphaFold2 generates a per-residue confidence metric called predicted local distance difference test (pLDDT), which ranges from 0 to 100. Based on the local distance difference test C α (lDDT-Cα) (Mariani et al., 2013), pLDDT can be utilized to approximate the concurrence between the prediction and the experimental structure’s agreement. The correlation between per-residue plDDT and the actual lDDT-C α is demonstrated in Fig. 5, with data based on 339 peptide chains in the benchmark dataset from this study, totaling 10,896 residues. The scatterplot demonstrates that there is a correlation between plDDT and lDDT-Cα (Pearson’s r = 0.60) and that the majority of the data points are distributed in [80–100], which is considered to be the region with high confidence in the AlphaFold2 predicted structure.

Figure 5 Correlation between per-residue pLDDT and lDDT-C α.

Data based on 339 peptide chains in the benchmark dataset, totaling 10,896 residues.

In addition, structure comparison diagrams of four AMPs are also shown in Fig. 6, and the bands of the predicted structures were colored using confidence. The consistency between the four predicted structures and the experimental structures is shownin Fig. S2, with the score of each residue using lDDT-Cα, which is mainly used to compare the local deviations between the structures. The structure comparison plots and consistency analyses show that the structures predicted by AlphaFold2 demonstrate a high degree of accuracy, sufficient for performing downstream analyses.

Figure 6 AlphaFold2 prediction and experimental structure for 4 AMPs.

(A) PDB 2LLD, RMSD = 0.767Å, (B) PDB 6QBK, RMSD = 0.853Å, (C) PDB 2L2R, RMSD = 0.624Å, (D) PDB 1M8A, RMSD = 0.334Å. The bands of the predicted structures are colored using confidence.

Performance of hyperparameters

The model’s hyperparameters are vital to the model’s performance. In this study, we used training dataset to compare different hyperparameters’ performance while applying 5-fold cross-validation. Mainly: the graph convolution algorithm, the number of residue-level features, the threshold of the contact map, and the protein language model.

In this study, the pre-trained Bi-LSTM protein language model was used to construct the node feature matrices. By adding output options, the Bi-LSTM was able to output different number of features: 21, 100, 2,069, 4,117, and 6,165. Three different graph convolution algorithms were used in deepAMPNet: GCN, GAT, and HGPSL. In order to control variables, the performance of the different algorithms was first compared by using the number of 21 and 100 residue features. As displayed in Fig. 7A, the accuracy when employing HGPSL surpasses that of both GCN and GAT, so the algorithm we used in subsequent comparisons was HGPSL.

Figure 7 Performance of different hyperparameters.

(A) Accuracy of different graph convolution algorithms, (B) Accuracy of different number of residue-level features, (C) Accuracy of different distance thresholds, (D) Accuracy of different protein language models.

Different number of residue-level features affects the final performance of deepAMPNet, we compared the following number of features: 21, 100, 2,069, 4,117, and 6,165. The final performance is demonstrated in Fig. 7B, the highest accuracy was achieved with the number of 6,165 features. As stated in the Bi-LSTM article, 6,165 residue-level features are more suitable for transfer learning. Therefore, in all subsequent experiments, the number of features encoded by Bi-LSTM was set to 6,165.

To define whether there is an interaction between two residues, we set a distance threshold for contact map construction. We compared the performance of deepAMPNet when 6 Å, 8 Å, 10 Å, 20 Å, and 30 Å were used as distance thresholds, respectively, and as shown in Fig. 7C, the highest accuracy was achieved when the distance threshold was 20 Å. Therefore, in the subsequent training of the deepAMPNet, we used a distance threshold of 20 Å to construct the contact maps.

In addition to Bi-STLM, we introduced two state-of-the-art protein language models ProtTrans (Elnaggar et al., 2021) and ESM2 (Lin et al., 2023), and compared the performance of these three protein language models with the three hyperparameters identified above. ProtTrans is a self-supervised pre-training language models was trained on data from UniRef and BFD containing up to 393 billion amino acids. Dimensionality reduction revealed that the ProtTrans-embeddings can capture biophysical features of protein sequences, and the embeddings demonstrated strengths in several biology tasks. ESM2 is a state-of-the-art general-purpose protein language model can be used to predict structure, function and other protein properties directly from individual sequences. As shown in Fig. 7D, we compared the performance of deepAMP when encoding residue-level features using each of these three language models separately. The number of residue-level features encoded by Bi-LSTM, ESM2, and ProtTrans are 6,165, 5,120, and 1,024, respectively. The accuracy of deepAMP when using Bi-LSTM embeddings outperformed ProtTrans and ESM2 by a narrow margin, perhaps because a larger number of features may contain additional information that can be used to distinguish AMPs from Non-AMPs. Based on the above results, we used Bi-LSTM for encoding residue-level features in deepAMPNet.

Based on the study in this section, we set the hyperparameter combination of deepAMPNet with HGPSL, 6,165 Bi-LSTM residue-level features, and a distance threshold of 20 Å to train the model and compared the performance on the independent test datasets.

Performance on the XUAMP dataset

Utilizing the XUAMP dataset proposed by Xu et al. (2021) we compared deepAMPNet with 11 state-of-the-art AMPs predictors including: amPEPpy (Lawrence et al., 2021), AMPfun (Chung et al., 2019), AMPEP (Bhadra et al., 2018), ADAM-HMM (Lee et al., 2015), ampir (Fingerhut et al., 2021), AMPScannerV2 (Veltri, Kamath & Shehu, 2018), AMPGram (Burdukiewicz et al., 2020), Deep-AMPEP30 (Yan et al., 2020), CAMP-ANN (Waghu et al., 2016), sAMPpred-GAT (Yan et al., 2023) and AMPpred-MFA (Li et al., 2023). To avoid overestimating the performance of deepAMPNet, sequences with similarity above 90% with the XUAMP dataset were removed from the deepAMPNet training dataset.

The comparison results of different predictors are displayed in Fig. 8, Table 3 and Fig. S3, from which we can observe that deepAMPNet achieved the highest performance with results as follows: area under the ROC curve (AUC), Acc, Mcc, sensitivity and F1-score. The results revealed that deepAMPNet achieved highest AUC with fewer false positives. As a result, deepAMPNet is sufficient for the identification of AMPs.

Figure 8 ROC curves of deepAMPNet and the other 11 state-of-the-art predictors on the XUAMP test dataset.

Table 3 The performance of deepAMPNet and the other 11 predictors on the independent test XUAMP dataset in terms of Acc, Mcc, Sn, Sp, Precision and F1-score.

The bold values show the best performance of deepAMPNet and the proposed predictors.

Method	Acc	Mcc	Sn	Sp	Precision	F1-score	
amPEPpy	0.679	0.432	0.400	0.958	0.906	0.555	
AMPfun	0.674	0.414	0.406	0.943	0.877	0.555	
AMPEP	0.661	0.429	0.330	0.992	0.975	0.493	
ADAM-HMM	0.684	0.390	0.521	0.847	0.773	0.623	
ampir	0.563	0.156	0.266	0.859	0.654	0.379	
AMPScannerV2	0.568	0.137	0.523	0.613	0.575	0.548	
AmpGram	0.564	0.131	0.445	0.682	0.584	0.505	
Deep-AMPEP30	0.533	0.183	0.065	1.000	1.000	0.122	
CAMP-ANN	0.584	0.182	0.385	0.782	0.639	0.481	
sAMPpred-GAT	0.712	0.456	0.529	0.895	0.835	0.647	
AMPpred-MFA	0.673	0.412	0.402	0.945	0.879	0.551	
deepAMPNet	0.758	0.531	0.639	0.876	0.838	0.725	

Performance on the MFA_test dataset

We comprehensively evaluated the performance of deepAMPNet on the MFA_test dataset with other 11 state-of-the-art AMPs predictors including: ampir (Fingerhut et al., 2021), CAMP3 (RF), CAMP3 (SVM), CAMP3 (DA), CAMP3 (ANN) (Waghu et al., 2016), iAMPpred (Meher et al., 2017), AMPScannerV2 (Veltri, Kamath & Shehu, 2018), AI4AMP (Lin et al., 2021), AMPlify (Li et al., 2022), AMPfun (Chung et al., 2019) and AMPpred-MFA (Li et al., 2023). We present ROC curves of all predictors in Fig. 9, where it can be observed that deepAMPNet achieved the highest performance of AUC. Additionally, we demonstrate the results of six metrics—Acc, Mcc, Sn, Sp, precision, and F1-score—in Fig. 10 and Table 4. These indicators highlighted that deepAMPNet led in performance across all six metrics. Among these predictors, deepAMPNet reached an accuracy of 95.2%, indicating that deepAMPNet is an effective predictor of AMPs.

Figure 9 ROC curves of deepAMPNet and the other 11 state-of-the-art predictors on the MFA_test dataset.

Figure 10 Results of six metrics of deepAMPNet and the other 11 predictors on the MFA_test dataset.

Table 4 The performance of deepAMPNet and the other 11 predictors on the independent MFA_test dataset in terms of Acc, Mcc, Sn, Sp, Precision and F1-score.

The bold values show the best performance of deepAMPNet and the proposed predictors.

Method	Acc	Mcc	Sn	Sp	Precision	F1-score	
ampir	0.550	0.143	0.208	0.896	0.670	0.318	
CAMP3(RF)	0.667	0.351	0.833	0.498	0.627	0.716	
CAMP3(SVM)	0.672	0.354	0.792	0.551	0.642	0.709	
CAMP3(DA)	0.680	0.364	0.766	0.592	0.656	0.707	
CAMP3(ANN)	0.757	0.514	0.751	0.762	0.762	0.757	
iAMPpred	0.766	0.535	0.828	0.703	0.739	0.781	
AMPScannerV2	0.781	0.578	0.904	0.655	0.727	0.806	
AI4AMP	0.784	0.568	0.787	0.781	0.784	0.786	
AMPlify	0.900	0.801	0.873	0.927	0.924	0.898	
AMPfun	0.922	0.845	0.937	0.908	0.911	0.924	
AMPpred-MFA	0.944	0.889	0.950	0.939	0.940	0.945	
deepAMPNet	0.952	0.905	0.966	0.939	0.941	0.953	

Discussion

In this study, we proposed deepAMPNet, an AMPs predictor employing the Bi-LSTM protein language model and the AlphaFold2 predicted structure. In a comparative performance assessment against other cutting-edge predictors, deepAMPNet achieved the highest or highly comparable performance in seven metrics. The exceptional performance of deepAMPNet can be attributed to the following innovative features:

(1) The use of Bi-LSTM model: Bi-LSTM protein language model utilized tens of millions of protein sequences from Uniref90, integrated PDB data for structure-supervised learning. This approach enabled the Bi-LSTM model to encode features that concurrently capture the sequence, structure, and evolutionary information of proteins. This information holds paramount importance in protein functional analysis, particularly in the identification of AMPs. Based on the properties of Bi-LSTM, the model integrates the context of amino acids during training. The global information of the sequence is integrated in the encoded features, which makes up for the lack of only local information in the existing model. Concurrently, we also evaluated the performance and effectiveness of different quantities of features to avoid potential redundancy or insufficiency of features.

(2) The use of protein structure: Protein sequence determines structure, and protein structure is the key to exercising its function. Protein secondary structures used only in existing models may not be sufficient to characterize AMPs. In this study, we firstly predicted the tertiary structure of AMPs through AlphaFold2. By setting a distance threshold, we effectively represented the structural features of proteins as computer-recognizable graph data. The experimental results showed that the accuracy reached 95.328% when using only structural information (one-hot+HGPSL) with 5-fold cross-validation on the training dataset (Fig. 7A).

(3) The use of HGPSL: In order to avoid the shortcomings of improperly selected neural networks in existing models, we used a graph neural network based on structure learning to represent protein structures. Hierarchical graph pooling with structure learning (HGPSL) can proficiently implement graph pooling processing to glean feature information of nodes from both the local and global structure of the graph. Further, this algorithm utilized the node features along with the total topological data of the graph to predict labels correlated with the graph. In this study, we compared the performance of GCN, GAT and HGPSL in AMPs identification. The results displayed in Fig. 7A showed that HGPSL outperformed the traditional graph convolution algorithms GCN and GAT in terms of accuracy when only a small number of features (one-hot) were used.

(4) Enhancement of training dataset: In this study, we collected AMPs from five comprehensive databases and retained 11,485 AMPs after removing redundancy. In comparison to other predictors that only have thousands of AMPs in the training dataset, the augmentation in our training dataset propounded an enhancement in deepAMPNet’s generalization capability.

Conclusions

In this study, we proposed an innovative antimicrobial peptide predictor named deepAMPNet, which utilizes protein sequences and AlphaFold2 predicted structures. deepAMPNet represents protein structures as computer-readable graph data by encoding node features using a pre-trained Bi-LSTM protein language model and constructing adjacency matrices based on a distance threshold. Following this, hierarchical graph pooling with structure learning-based (HGPSL-based) is used to learn antimicrobial peptide features from the graph data, which is ultimately input into the fully connected layer for antimicrobial peptide identification. Experimental results conducted with other state-of-the-art predictors on the latest independent test datasets show that deepAMPNet achieved the highest or highly comparable performance in seven metrics. Therefore, we can conclude that deepAMPNet can facilitate enhanced accuracy and proficiency in antimicrobial peptide identification, thereby aiding researchers to devise pertinent drugs to counter the escalating issue of antimicrobial resistance.

Moreover, considering the escalating count of structures in AlphaFold protein structure database alongside the superior performance of the Bi-LSTM protein language model in downstream transfer learning, it is reasonable to presume that the protein recognition model, built on the predicted structure and the Bi-LSTM model, possesses wide-ranging applicability. It also implies that other functional peptides, including bioactive peptides like antioxidant peptides and immunological peptides, can likewise exhibit exceptional performance.

Supplemental Information

Supplemental Information 1 Amino acid distribution of AMPs and Non-AMPs of training dataset

Supplemental Information 2 The consistency between the predicted structures and the experimental structures of 4 AMPs

Supplemental Information 3 Results of six metrics of deepAMPNet and the other 11 predictors on the XUAMP dataset

Additional Information and Declarations

Competing Interests

Author Contributions

Data Availability

The authors declare there are no competing interests.

Fei Zhao conceived and designed the experiments, performed the experiments, analyzed the data, prepared figures and/or tables, authored or reviewed drafts of the article, and approved the final draft.

Junhui Qiu performed the experiments, analyzed the data, prepared figures and/or tables, and approved the final draft.

Dongyou Xiang analyzed the data, prepared figures and/or tables, and approved the final draft.

Pengrui Jiao analyzed the data, prepared figures and/or tables, and approved the final draft.

Yu Cao performed the experiments, authored or reviewed drafts of the article, and approved the final draft.

Qingrui Xu performed the experiments, authored or reviewed drafts of the article, and approved the final draft.

Dairong Qiao conceived and designed the experiments, authored or reviewed drafts of the article, and approved the final draft.

Hui Xu conceived and designed the experiments, authored or reviewed drafts of the article, and approved the final draft.

Yi Cao conceived and designed the experiments, authored or reviewed drafts of the article, and approved the final draft.

The following information was supplied regarding data availability:

The code and data are available at GitHub and Zenodo:

– https://github.com/Iseeu233/deepAMPNet

– Iseeu233. (2024). Iseeu233/deepAMPNet: First release of deepAMPNet. (v1.0.0). Zenodo. https://doi.org/10.5281/zenodo.12527166.

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
