# Peer review of "deepAMPNet: a novel antimicrobial peptide predictor employing AlphaFold2 predicted structures and a bi-directional long short-term memory protein language model"

_PeerJ, doi:10.7717/peerj.17729_

## Round 0.1 · original submission · Minor Revisions

· Academic Editor

Minor Revisions

Please address the comments and questions by reviewers.

Reviewer 1 ·

Basic reporting

This manuscript introduces deepAMPNet, a new antimicrobial peptide predictor that combines AlphaFold2 and a Bi-directional LSTM protein language model. It uses Bi-directional LSTM and one-hot encoding to generate embeddings for each of the 21 amino acid residues. These embeddings, along with the contact map generated from AlphaFold2, are used as input features to a GNN structure for AMP classification. The authors also conducted a comparison analysis with other state-of-the-art AMP predictors based on two external independent datasets.

In summary, this paper is clearly written and understandable. It has complete experiments to demonstrate the performance of deepAMPNet in AMP prediction. I have a few concerns or comments that need to be addressed by the authors.

General comments:
1. The authors utilized Bi-LSTM to encode residue-level features. If I understand correctly, the output of Bi-LSTM is an embedding for a given protein sequence instead of embedding sets for each of the unique residues in the protein. Therefore, the main difference in input features across the residues (nodes) in the contact-map-based graph is the first 21 one-hot embeddings. If that is the case, the Bi-LSTM embedding might not provide more help than simple embeddings as input for GNN in distinguishing the residuals. So, I am wondering why we can’t just use one-hot embedding to HGPSL to generate embedding and then combine it with the Bi-LSTM to do the final prediction. The author can consider this structure and evaluate its performance.
2. Except for Bi-STLM, have the authors tried other SOTA protein models such as ProTrans (https://arxiv.org/abs/2007.06225). I think the authors can provide further comparisons to demonstrate the effectiveness of Bi-STLM.

Minor comments:
1. In abstract: “have emerged as a potent solution AMR ….” -> “have emerged as a potent solution to AMR ….”
2. Lines 108-110: the sentence “ In existing methods, present an insufficiency or redundancy in terms of number of features.” is not complete.
3. Lines 154 - 155: For description “those lacking non-canonical residues like 'B', 'J', 'O', 'U', 'X' or 'Z', whose structures aren't predictable in AlphaFold2”, I am confused about if those AMPs lacking non-canonical residues are removed or retained. Please modify this sentence to make it clear.

Experimental design

no comment

Validity of the findings

no comment

Reviewer 2 ·

Basic reporting

The manuscript used clear and unambiguous professional English. Literature references and sufficient field context were provided. The article structure is good.

Experimental design

There are three hyperparameters in your approach. Are u using cross-validation to select the values for the three hyperparameters simultaneously? If not, how did u select each hyperparameter respectively with the control of other hyperparameters? How did you decide the candidates of the numbers of residue-level features and the numbers of distance thresholds? I have a concern that the model has overfitting issue.

Validity of the findings

The background of this study is about the issue of antimicrobial resistance (AMR) threatens global public health, with antimicrobial peptides (AMPs) emerging as a promising solution due to their therapeutic potential. This study introduces deepAMPNet, a model utilizing graph neural networks, AlphaFold2-predicted structures, Bi-LSTM protein language model, and adjacency matrices to swiftly identify AMPs. Compared to other predictors, deepAMPNet demonstrates superior accuracy and performance across various evaluation metrics. Overall, deepAMPNet integrates structural and sequence information to advance antimicrobial peptide pharmaceuticals. The findings will be relatively promising if the authors can resolve the concern about hyperparameter selection.

Additional comments

NA

Reviewer 3 ·

Basic reporting

All comments have been added in detail to the 4th section called additional comments.

Experimental design

All comments have been added in detail to the 4th section called additional comments.

Validity of the findings

All comments have been added in detail to the 4th section called additional comments.

Additional comments

Review Report for PeerJ Computer Science
(deepAMPNet: a novel antimicrobial peptide predictor employing AlphaFold2 predicted structures and a Bi-directional Long Short-Term Memory protein language model)

1. Within the scope of the study, a model called deepAMPNet was developed for antimicrobial peptide predictor using deep learning-based graph neural networks and Bi-directional Long Short-Term Memory.

2. More than one different dataset was used as dataset in the study, and this is very important and valuable for the application of the developed model.

3. The codes and datasets related to the application part of the study have been shared on the github platform, which is positive in terms of the controllability of the study and its possible contributions to the post-publication literature.

4. Although the introduction section mainly covers the literature, it is recommended that in this section, first a literature table and then the study's differences from the literature and its contributions to the literature should be emphasized.

5. Cross-validation is frequently used in dataset distribution to analyze the results in classification problems. Please interpret the study from this perspective.

6. When the study is examined in terms of evaluation metric, it is observed that most important metrics are obtained. In this respect, both the metric type and the results obtained are at an acceptable level.

7. In addition to the deep learning model developed in the study, other existing models in the literature have been used, which increases the confidence in the model result.

As a result, when the model developed within the scope of the study is examined, the study has unique points. However, it is recommended to pay attention to the above-mentioned parts in order to increase the post-publication citation potential of the study and its contribution to the literature.

---

## Round 0.2 · accepted · Accept

· Academic Editor

Accept

Your manuscript has been accepted for publication. Congratulations!

Reviewer 2 ·

Basic reporting

The refined paper has a good format and structure of figures, tables, and paragraphs.

Experimental design

Since the authors carefully addressed the issues pointed out by the reviewers. The experimental design is promising.

Validity of the findings

The refined paper presented results and conclusions clearly. Since the authors addressed the issues and concerns in comments, the findings are promising.

Additional comments

NA

Reviewer 3 ·

Basic reporting

All comments have been added in detail to the last section.

Experimental design

All comments have been added in detail to the last section.

Validity of the findings

All comments have been added in detail to the last section.

Additional comments

Review Report for PeerJ
(deepAMPNet: a novel antimicrobial peptide predictor employing AlphaFold2 predicted structures and a Bi-directional Long Short-Term Memory protein language model)

Thanks for the revision. I have examined in detail the responses to the reviewer comments and this latest revised version of the paper. I believe that the answers are sufficient. When the subject discussed in the study is examined in terms of its importance, contribution to the literature and originality; I recommend that this paper be accepted without further revision. I wish the authors success in their future work. Kind regards.